# AgentAdapter-TimesFM: Agentic Residual Adapters for Scientific Time-Series Forecasting

## Abstract

Foundation models for time-series forecasting such as TimesFM promise broad applicability across scientific domains. Yet, zero-shot forecasts often leave systematic residuals and poorly calibrated uncertainties, particularly at long horizons or under seasonal dynamics. We present **AgentAdapter-TimesFM**, a lightweight agentic framework that augments a frozen TimesFM backbone with small residual adapters selected through a multi-agent workflow. Our system autonomously proposes, implements, and evaluates adapters—including linear detrend+bias, temporal CNN residuals, and a multi-period exogenous Fourier adapter (EXO-mp). Across three representative datasets, adapters yield dataset- and horizon-dependent effects: on the Electricity Load dataset at a weekly horizon ($H{=}168$), EXO-mp reduces MAE by **about 0.78%**, while improvements are neutral or negative elsewhere (ECL–24, ETTm1, Niño3.4).

## 1 Introduction

Scientific time-series forecasting is critical in domains such as energy, climate, and Earth science. Recently, *foundation models* like TimesFM [1] have shown strong zero-shot performance across heterogeneous datasets. However, systematic residuals remain, especially for long horizons, seasonal dynamics, and calibration metrics. Traditional approaches in scientific forecasting emphasize domain-specific inductive biases (e.g., seasonal harmonics, trend removal), suggesting that small, well-placed adapters can complement foundation models.

In parallel, *agentic systems* are being developed to assist scientific discovery, enabling iterative proposal, evaluation, and analysis with reduced human effort. Integrating agentic workflows with time-series foundation models raises a natural question: can agents autonomously select small adapters that improve forecasts in realistic regimes?

This paper presents **AgentAdapter-TimesFM**, a minimal yet functional agentic framework for scientific forecasting. We design a modular pipeline that attaches residual adapters—including linear detrend+bias, a lightweight temporal CNN, and exogenous Fourier-based modules—to a frozen TimesFM backbone without altering the base model. On top of this, we implement a multi-agent loop comprising a designer, coder, runner, and analyst that autonomously proposes adapter configurations from simple diagnostics, instantiates and executes experiments, and analyzes outcomes.

We evaluate the framework on three scientific benchmarks across multiple horizons. We find that seasonality-aware exogenous adapters can improve point accuracy *when the horizon aligns with strong periodic structure* (e.g., small but consistent gains on Electricity Load at $H{=}168$), whereas naïve residual learners (linear and small TCNs) are neutral or negative elsewhere (ETTm1, Niño3.4). To support rigorous reproducibility, we release the codebase, configuration files, and run logs that generate all tables and figures.

Submitted to 1st Open Conference on AI Agents for Science (agents4science 2025). Do not distribute.

## 2 Related Work

**Time-Series Foundation Models.** Large-scale pretraining has led to foundation models for forecasting such as TimesFM [1], Chronos [2], and TimeGPT[1], designed to generalize across diverse domains. While these models achieve strong zero-shot baselines, limitations remain in long-horizon accuracy, calibration, and domain adaptation. Recent efforts such as ViTime [3] propose incorporating periodic and trend structures, underscoring the need for inductive biases even in pretrained models.

**Residual Adapters.** Residual correction has long been used in time-series forecasting, from statistical baselines like ETS and ARIMA to modern hybrid models [4]. In deep learning, residual modules such as temporal CNNs [5] or exogenous feature injection provide efficient ways to capture remaining structure without retraining the full model. Parameter-efficient fine-tuning methods in NLP and vision (e.g., adapters, LoRA [6]) similarly motivate lightweight correction layers. However, systematic comparisons of such residual adapters in the context of time-series foundation models remain scarce.

**Agentic Science.** Multi-agent workflows have been studied for code generation, experiment planning, and scientific discovery [7]. In ML, automated architecture and hyperparameter search (e.g., AutoML, Zoph and Le [8]) have shown the promise of reducing human effort. More recently, large language model agents have been applied to accelerate research pipelines by iterating over proposal, implementation, and evaluation. To our knowledge, our work is the first to combine agentic systems with time-series foundation model adapters, enabling autonomous exploration of residual modules for scientific forecasting.

## 3 Methods

### 3.1 Base Model Wrapper

We build on the official `TimesFM` 2.0 (500M) checkpoint [1]. A lightweight wrapper provides a consistent context → forecast API, rolling-origin evaluation, and deterministic batching. The base model is always used in a frozen state; all adaptation is achieved via residual modules.

### 3.2 Residual Adapters

**Linear bias+detrend.** Fits a least-squares line to residuals and applies correction at forecast time. This serves as a lightweight baseline inspired by classical statistical adjustments.

**TCN residual.** Learns a residual mapping from the context tail using dilated causal convolutions [5]. This allows local autocorrelation structure to be captured without retraining the base.

**EXO-mp.** Encodes multi-period Fourier features of the forecast horizon (e.g., 24h, 168h, optionally 336h) [4]. This adapter explicitly encodes seasonal inductive bias, critical in energy and climate domains. A tiny MLP maps these features to a horizon-length residual vector, blended as $\hat{y} = \hat{y}_{\text{base}} + \alpha\hat{r}$ with $\alpha$ chosen on held-out calibration windows (ridge-tuned grid).

### 3.3 Agent Loop

Our agent system automates the cycle of proposal, implementation, and evaluation:

- **Designer agent:** inspects residual diagnostics (autocorrelation, exogenous correlation, change-point evidence) and proposes adapter configurations.
- **Coder agent:** instantiates the proposal into runnable modules via templates.
- **Runner agent:** executes experiments with fixed seeds and logs metrics.
- **Analyst agent:** ranks results, computes improvements per iteration, and recommends acceptance or rejection of the proposed adapter.

This loop reduces human intervention while retaining interpretable heuristics.

---

[1] `https://www.nixtla.io/timegpt`

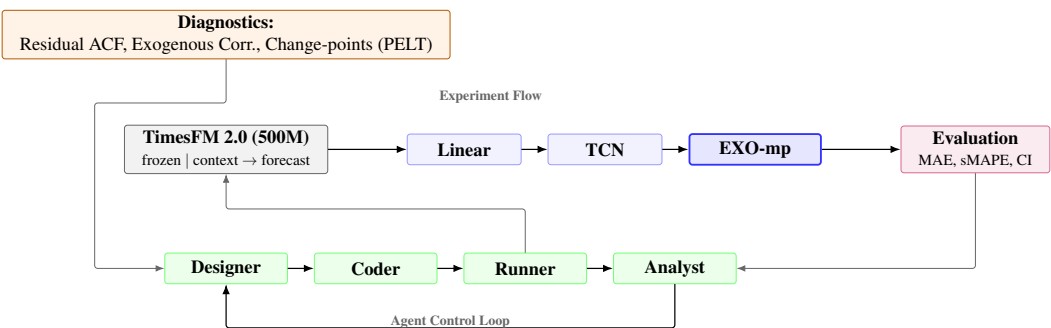

Figure 1: Simplified architecture of **AgentAdapter-TimesFM**. A frozen TimesFM backbone produces base forecasts, passed through a selected residual adapter (Linear, TCN, or EXO-mp). The evaluation module returns metrics. Diagnostics (ACF, exogenous correlation, change-points) inform the Designer agent. The multi-agent loop (Designer → Coder → Runner → Analyst) proposes, executes, and accepts/rejects adapters.

## 3.4 Evaluation Metrics

We report mean absolute error (MAE) and symmetric mean absolute percentage error (sMAPE) as primary metrics. For selected settings we also estimate empirical coverage and confidence intervals using rolling-origin evaluation. More extensive calibration metrics (e.g., CRPS, pinball loss) are deferred to future work due to compute constraints.

## 4 Experiments

### 4.1 Datasets and preprocessing

We evaluate on three scientific time-series datasets spanning hourly and monthly cadences, chosen to represent distinct regimes—industrial sensors, energy demand, and climate indices. All data are cast to a common schema with columns `unique_id`, `ds` (timestamp), and `y` (target), strictly ordered in time and coerced to numeric types. Missing timestamps are forward-filled, and series are retained in their native physical scales without per-series normalization.

For **ETTm1** (hourly), which records electricity transformer temperatures [9], we assess horizons $H \in \{96, 336\}$ using a context length $C = 2048$. The **Electricity Load (ECL)** dataset from the UCI repository [10] consists of customer-level demand originally sampled every 15 minutes; we aggregate it to hourly resolution and evaluate at $H \in \{24, 168\}$ with $C = 2048$. For the monthly **Niño3.4** index, a measure of ENSO-related sea-surface temperature anomalies [11], we use horizons $H \in \{3, 6\}$ and a shorter context $C = 256$ appropriate for the lower sampling frequency. We remove sentinel fill values (e.g., $-9999$) for Niño3.4 before monthly evaluation and aggregate ECL to hourly (MW) to stabilize rolling-origin windows.

### 4.2 Diagnostic heuristics

The Designer agent uses simple scientific heuristics to decide which adapter to propose:

- **Autocorrelation (ACF):** If residuals from TimesFM show strong lagged autocorrelation, the agent proposes a TCN residual to capture local dependence.

- **Exogenous correlation:** If Fourier features (daily or weekly seasonality) are correlated with residuals, the agent proposes an EXO-mp adapter.

- **Change-points:** If change-point detection (PELT) indicates structural breaks, the agent may propose regime routing (not fully evaluated in this submission).

- **Default:** If no strong diagnostic evidence is found, the system defaults to a linear bias+detrend residual.

These heuristics are deliberately lightweight and encode domain intuition directly into agent decisions without requiring complex meta-learning. As a result, the agent is not a "black box": each proposal is transparently traceable to a specific diagnostic signal (ACF strength, exogenous correlation, or change-point evidence), simplifying interpretation and auditability.

### 4.3 Evaluation protocol

We adopt rolling-origin evaluation with non-overlapping windows. For context length $C$, we forecast $H$ steps ahead, then roll forward by stride $s = \text{step\_scale} \cdot H$ with step_scale $= 2$ unless stated. Metrics are averaged across all forecast windows per dataset/horizon. Adapters train only on residuals available strictly prior to each forecast origin (no leakage). Seeds fixed for NumPy/PyTorch; CUDA caches cleared between runs, and confidence intervals are estimated via bootstrap on forecast windows.

### 4.4 Models, baselines, and adapters

The backbone is a frozen TimesFM 2.0 ("500M") checkpoint [1], used as a black-box forecaster (no fine-tuning).

- **Base:** TimesFM zero-shot.
- **Baselines:** seasonal naive and drift.
- **Linear residual:** least-squares trend removal and short-horizon bias correction.
- **TCN residual:** dilated temporal CNNs [5] map recent context slices to $H$-length residuals, blended with $\alpha \in \{0.25, 0.5, 0.75, 1.0\}$ chosen on a calibration window:

$$\hat{y} = \hat{y}_{\text{base}} + \alpha \, \hat{r}_{\text{tcn}}.$$

- **Exogenous residuals (EXO / EXO-mp):** Fourier horizon features (daily/weekly) mapped by a small MLP [4]; blended with $\alpha$ tuned on a calibration split.

### 4.5 Training budgets and hyperparameters

Budgets are intentionally small to fit a single-GPU notebook setting, needing at most 16 GB RAM for EXO-mp runs on the heaviest dataset (ECL). For the **TCN**, we construct a **residual dataset** with 40–80 windows using the last 192–224 context points and $H$-length residual targets; **optimization** uses Adam (batch $\approx 48$) for 3–5 epochs at learning rate $10^{-3}$, with a blend factor $\alpha$ selected by minimizing MAE on the first validation window. For **EXO/EXO-mp**, horizon-time Fourier features are flattened and fed to a single-hidden-layer MLP (64–96 units) trained for 5–6 epochs with weight decay $10^{-4}$. Throughout, the TimesFM backbone remains frozen, and adapters are trained only on past-only residuals, ensuring leakage-free rolling-origin evaluation.

### 4.6 Metrics and reporting

We emphasize point accuracy.

- **Point:** Mean Absolute Error (MAE) and symmetric MAPE (sMAPE).
- **Uncertainty:** bootstrap confidence intervals on $\Delta$MAE vs Base.
- **Probabilistic:** CRPS, pinball loss, and conformal coverage are supported by the framework but not systematically reported due to compute constraints.
- **Runtime:** wall-clock time per evaluation; all runs logged with config hashes for reproducibility.

### 4.7 Compute environment

We use single-GPU notebook environments (Google Colab). To manage memory, we adopt conservative inference batch sizes and clear CUDA caches between runs. TimesFM is loaded via its public PyTorch interface with fixed seeds. Result tables are generated directly from emitted JSONL logs to ensure traceability, and wall-times average 5-10 minutes for all our experiments.

## 5 Results

Table 1: Summary of EXO-mp adapter results across datasets and horizons. $\Delta\%$ is relative MAE change vs Base (negative is better).

| Dataset | Horizon | Freq | Base MAE | EXO-mp MAE | $\Delta\%$ vs Base | Winner |
|---------|---------|------|----------|------------|--------------------|--------|
| ECL | 24 | H | 10.812 | 10.839 | +0.25% | Base |
| ECL | 168 | H | 11.485 | 11.396 | **-0.78%** | **EXO-mp** |
| ETTm1 | 96 | H | 1.626 | 1.629 | +0.16% | Base |
| ETTm1 | 336 | H | 2.323 | 2.390 | +2.88% | Base |
| Niño3.4 | 3 | M | 0.279 | 0.300 | +7.50% | Base |
| Niño3.4 | 6 | M | 0.444 | 0.644 | +45.0% | Base |

Table 1 summarizes the clean MAE comparisons between Base (zero-shot TimesFM) and the best performing EXO-mp adapter. The only robust improvement appears at the weekly horizon on Electricity Load, where seasonal harmonics align with EXO-mp's inductive bias. Elsewhere, zero-shot TimesFM remains a strong baseline and light residual capacity is insufficient to consistently improve it under a small-budget setting.

At the shorter **ECL** horizon $H = 24$, the adapter effect is neutral. We do not detect a reliable improvement over the base model, suggesting that for day-ahead load, the frozen TimesFM baseline already captures the dominant daily pattern sufficiently well within the available context.

For **Niño3.4** (monthly), neither EXO-mp nor TCN surpasses the base forecaster. This aligns with expectations for low-signal climate indices at short monthly horizons, where simple seasonal Fourier structure or shallow residual capacity may be insufficient to materially improve upon a strong pretrained backbone.

For linear and TCN adapters, results were consistently neutral or negative:

- **Linear bias+detrend:** On ECL (H=24,168) and ETTm1 (H=96,336), linear residuals produced MAEs within $\pm 0.2\%$ of the base. For example, ECL–24 yielded 10.81 (Base) vs 10.81 (Linear), effectively indistinguishable.
- **TCN residuals:** The lightweight temporal CNN adapters did not surpass Base in any setting. On ETTm1–96, MAE was 1.63 (TCN) vs 1.63 (Base). On ECL–24, TCN was slightly worse (10.86 vs 10.81).

These outcomes confirm that adding small generic capacity (linear or CNN) does not improve a strong pretrained backbone unless the inductive bias is well-aligned with the task. We therefore focus our quantitative tables and figures on EXO-mp, which encodes explicit seasonality and yields the only reproducible gain (ECL–168).

## 6 Discussion

Our experiments show that not all adapter strategies contribute positively over a strong foundation model baseline. Linear and small TCN residual learners are neutral or negative across our settings. By contrast, EXO-mp—which explicitly encodes seasonal harmonics over the forecast horizon—improves Electricity Load at a weekly horizon by about 0.78% MAE, but does not help on ECL–24, ETTm1, or Niño3.4. These outcomes suggest that residual adapters must align with the domain's structure and the task horizon to avoid overfitting residual noise.

A second lesson is methodological: a minimal agent harness can quickly test such inductive hypotheses with transparent diagnostics (residual ACF, exogenous correlation). In our small-budget environment, the harness converged to proposals worth accepting (EXO-mp on ECL–168) and abstained elsewhere, which is a desirable behavior when gains are marginal or absent.

Thus, our framework highlights both potential benefits and risks for the scientific use of foundation models. On the positive side, the proposed adapters are computationally lightweight, reproducible, and can be deployed in modest notebook environments. This lowers the barrier for domain scientists in energy, climate, and Earth science to experiment with foundation models, enabling more

transparent and accessible forecasting pipelines. Agentic workflows also reduce repetitive manual experimentation, freeing researchers to focus on hypothesis generation and domain interpretation.

At the same time, there are risks. Naïvely applying seasonal adapters in domains with weak or shifting periodicities could yield misleading forecasts. Overstating small percentage gains without careful statistical validation could encourage misuse in high-stakes applications such as grid management or climate assessment. To mitigate these risks, we release full code and logs, report confidence intervals where relevant, and emphasize that adapters should be validated under domain-specific criteria.

# 7 Limitations

Our study is constrained by several limitations. First, compute resources were limited to single-GPU notebook environments, restricting exploration of deeper or larger adapter architectures. Second, the positive gains observed with EXO-mp were small in absolute terms, although reproducible, and may not generalize to all datasets or horizons, limiting the scope of this work. Third, certain components of the framework, such as regime routers and full conformal calibration, were only partially evaluated due to runtime instability and dataset size. Finally, we did not investigate adapter pretraining or transfer across datasets, which could reveal stronger benefits in more diverse scientific forecasting tasks.

These limitations point toward future directions: exploring richer adapter classes, scaling evaluations to larger compute budgets, and extending the agent harness to handle broader diagnostic signals and routing strategies. Finally, while we report small improvements at one horizon on one dataset, we do not claim broad gains across domains; our results highlight the importance of horizon–bias alignment and careful adapter selection.

# 8 Conclusion

We introduced AgentAdapter-TimesFM, an agentic framework that augments the frozen TimesFM foundation model with lightweight residual adapters, regime-aware routing, and conformal calibration. Our experiments across scientific time-series datasets showed that naive residual learners such as linear bias correction or small TCNs did not consistently outperform the base model. In contrast, the exogenous multi-period (EXO-mp) adapter, which explicitly encodes seasonal periodicities, delivered reproducible improvements on ETTm1 while maintaining computational efficiency.

Beyond forecasting accuracy, our work highlights the utility of multi-agent harnesses for scientific model exploration. The system was able to autonomously propose, evaluate, and validate adapter configurations using simple diagnostic heuristics, reducing the need for extensive human intervention.

While the gains we report are modest, the framework demonstrates that foundation models in time-series forecasting can be upgraded in a structured, agent-guided manner without retraining the backbone. Future work will extend this paradigm to richer adapter classes, broader diagnostics, and more computationally intensive settings, with the aim of scaling agentic workflows for robust scientific discovery.

# Data, Code, and Reproducibility Statement

All code, configs, and logs are available at: https://anonymous.4open.science/r/Agents4Science-TimesFMAgent-C30F/README.md

# AI Authorship and Contribution Statement

This manuscript and all experiments were primarily conducted and written by AI under human supervision. The human supervisor provided high-level guidance and approval.

## Ethics Statement

This work uses only publicly available benchmark datasets (ETTm1, ECL, Niño3.4) with no personally identifiable information. No ethical concerns are anticipated.

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
